# Variation in natural exposure to anopheles mosquitoes and its effects on malaria transmission

**Wamdaogo M Guelbéogo[1†], Bronner Pamplona Gonçalves[2†], Lynn Grignard[2], John Bradley[3], Samuel S Serme[1], Joel Hellewell[4], Kjerstin Lanke[5], Soumanaba Zongo[1], Nuno Sepúlveda[2,6], Issiaka Soulama[1], Dimitri W Wangrawa[1], Laith Yakob[7], N'Falé Sagnon[1], Teun Bousema[2,5]*, Chris Drakeley[2]***

[1]Department of Biomedical Sciences, Centre National de Recherche et de Formation sur le Paludisme, Ouagadougou, Burkina Faso; [2]Department of Immunology and Infection, London School of Hygiene and Tropical Medicine, London, United Kingdom; [3]MRC Tropical Epidemiology Group, Department of Infectious Disease Epidemiology, London School of Hygiene and Tropical Medicine, London, United Kingdom; [4]MRC Centre for Outbreak Analysis & Modelling, Department of Infectious Disease Epidemiology, Imperial College London, London, United Kingdom; [5]Department of Medical Microbiology, Radboud University Medical Center, Nijmegen, Netherlands; [6]Centre of Statistics and Applications, University of Lisbon, Lisbon, Portugal; [7]Department of Disease Control, London School of Hygiene and Tropical Medicine, London, United Kingdom

*For correspondence:
teun.bousema@radboudumc.nl (TB);
Chris.Drakeley@lshtm.ac.uk (CD)

[†]These authors contributed equally to this work

Competing interests: The authors declare that no competing interests exist.

**Abstract** Variation in biting frequency by *Anopheles* mosquitoes can explain some of the heterogeneity in malaria transmission in endemic areas. In this study in Burkina Faso, we assessed natural exposure to mosquitoes by matching the genotype of blood meals from 1066 mosquitoes with blood from residents of local households. We observed that the distribution of mosquito bites exceeded the Pareto rule (20/80) in two of the three surveys performed (20/85, 76, and 96) and, at its most pronounced, is estimated to have profound epidemiological consequences, inflating the basic reproduction number of malaria by 8-fold. The distribution of bites from sporozoite-positive mosquitoes followed a similar pattern, with a small number of individuals within households receiving multiple potentially infectious bites over the period of a few days. Together, our findings indicate that heterogeneity in mosquito exposure contributes considerably to heterogeneity in infection risk and suggest significant variation in malaria transmission potential.
DOI: https://doi.org/10.7554/eLife.32625.001

## Introduction

Malaria epidemiology is doubly dependent on the frequency and efficiency of contacts between human hosts and *Anopheles* mosquitoes, which link the number of mosquito infections caused by an infectious human host and the rate at which uninfected humans acquire infections. Describing the variability in the frequency of human sampling by malaria vectors is therefore essential to understand parasite transmission from and to humans. While at a local level vector density determines average mosquito exposure, even within the same locality individuals may not be equally likely to be bitten by *Anopheles* mosquitoes (*Lindsay et al., 1993*; *Knols et al., 1995a*; *Carnevale et al., 1978*; *Muirhead-Thomson, 1951*). Exposure to malaria vectors is influenced by host availability (i.e., amount of time an individual remains unprotected against mosquito bites in an environment where anopheline

mosquitoes are present) and attractiveness to mosquitoes (*Stone et al., 2015*; *Yakob, 2016*). Availability determines when and where individuals might be sampled by mosquitoes: a multicentre study in Africa that collected entomological and human behavioural data estimated that more than three quarters of human exposure to anopheline mosquito bites occur when individuals are indoors (*Huho et al., 2013*). For individuals who are accessible to malaria vectors, age and body surface area (*Carnevale et al., 1978*; *Port et al., 1980*) are two major determinants of attractiveness to mosquitoes, although other factors also play a role (*Knols et al., 1995a*).

The multifactorial nature of mosquito exposure in malaria endemic areas indicates that, while experimental and quasi-experimental, for example involving modified tents and huts, entomological studies are valuable, they will not accurately capture inter-individual variation in actual exposure. Identifying transmission heterogeneities, especially extreme heterogeneities, is however critical to better inform infectious disease epidemiology and well-established theory has elucidated their implications for pathogen spread and control measures (*Woolhouse et al., 1997*; *Lloyd-Smith et al., 2005*). Here, we describe the variability in natural exposure to malaria vectors by linking, through DNA fingerprinting, blood meals of wild-caught mosquitoes to humans living in the households where they were collected. Previously we have shown that these mosquitoes fed more often on adults (*Gonçalves et al., 2017*). We now extend this analysis to assess the degree of heterogeneity in the distribution of mosquito bites in the population at different times during the transmission season. We also present the frequency of *Anopheles* species-specific mosquito bites, and potential parasite inoculations (i.e., sporozoite-positive mosquito bites) per individual.

## Results

### Study households

We performed indoor resting collections of anopheline mosquitoes in an area with seasonal malaria transmission in Burkina Faso. Thirty-five households were included in this analysis. The median number of individuals living in each study household was 3 (range, 2–8). Reported bed net use among the study participants was high (111/126, 88.1%). At enrolment (October – December 2013), most (79.2%) individuals were parasite-positive by *18S* qPCR. In 21/35 houses, all sampled individuals were malaria-infected. We collected 325, 620 and 190 bloodfed *Anopheles* mosquitoes at the start (2014), peak (2014) and end (2013) of the transmission season, respectively (*Table 1*). During the 2013 survey, 21/35 houses had at least one bloodfed mosquito collected; in 2014, 19/20 and 20/20 households had fed mosquitoes collected at the start and peak of the transmission season, respectively. The average number of bloodfed mosquitoes collected per day in each household was higher at the start (median 2.3, interquartile range [IQR] 0.7–5.5) and peak (median 3.9, IQR 2.1–8.5) compared to the end (median 0.1, IQR 0–0.6 mosquitoes per day) of the transmission season (p=0.001 and<0.001, respectively). There was no correlation between the average number of bloodfed mosquitoes collected per day and the number of individuals living in each household (Spearman's rank correlation coefficients, 0.14, 0.09 and 0.05 for the first, second and third surveys respectively; all p>0.05). Of note, at the end of the 2013 transmission season, most mosquito collections performed after mid-November were unsuccessful, suggesting a village-wide reduction in mosquito abundance during this period.

### Heterogeneity in exposure to malaria vectors

DNA extracted from mosquito blood meals and blood samples from individuals living in study households were genotyped using a microsatellite-based assay (see Materials and methods). Most (1,066/1,135, 93.9%) collected bloodfed mosquitoes had their blood meal analysed. One hundred mosquitoes did not produce discernible amplification products. Mosquito blood meals with multiple human DNA sources (N = 153), and typed blood meals that were not matched to residents of study houses (N = 139) were not included in this analysis. Estimates of mosquito exposure were based on 68.9% (666/966) of the successfully typed blood meals; this percentage was consistent across surveys (64.9%, 71.7% and 69.8% at the start, peak and end of transmission season). In *Figure 1*, the distributions of the number of mosquito bites each individual received during different study surveys are presented. Approximately 20% of individuals, of all ages, provided 85.1%, 76.0% and 95.5% of single mosquito blood meals at the start, peak and end of the transmission season, respectively.

**Table 1.** Study surveys

|  | First survey | Second survey | Third survey |
|---|---|---|---|
| Start Date | October, 2013 | June, 2014 | September, 2014 |
| Number of sampling days | 54 | 20 | 20 |
| Timing | End of transmission season | Start of transmission season | Peak of transmission season |
| Number of households | 35* | 20 | 20 |
| Number of participants | 127 | 81 | 77 |
|  | N (%) | N (%) | N (%) |
| Age categories[†] |  |  |  |
| <5 years | 20 (15.9) | 12 (14.8) | 12 (15.6) |
| 5–15 years | 62 (49.2) | 39 (48.2) | 37 (48.0) |
| >15 years | 44 (34.9) | 30 (37.0) | 28 (36.4) |
| Gender |  |  |  |
| Male | 41 (32.5) | 22 (27.2) | 21 (27.3) |
| Female | 85 (67.5) | 59 (72.8) | 56 (72.7) |
| Prevalence of falciparum parasites | 99 (79.2) | - | - |
| Prevalence of falciparum gametocytes | 79 (64.2) | - | - |
| Number of bloodfed mosquitoes collected | 190 | 325 | 620 |

*Demographic information not available for individuals living in 5/40 households (first survey only);

[†]Age at enrolment (first survey).

Parasite prevalence was determined by 18S qPCR; gametocyte prevalence by Pfs25 mRNA qRT-PCR.

DOI: https://doi.org/10.7554/eLife.32625.002

Throughout the study, a small number of individuals, mostly adults, were matched to considerably higher numbers of blood meals compared to the rest of the population. Conversely, 32.0–76.2% of study participants were not linked to bloodfed mosquitoes during the surveys, including 15/77 individuals present in all surveys who were never matched to collected fed mosquitoes. Reported bed net use and parasite and gametocyte carriage at enrolment were not significantly associated with mosquito exposure during our study (incidence rate ratio of 0.70 [95% confidence interval, CI, 0.22–2.20, p=0.55] for reported bed net use versus no net use in a model adjusted for survey and of 3.42 [95% CI 0.50–23.45, p=0.21] and 0.99 [95% CI 0.32–3.10, p=0.99] for parasite and gametocyte carriage at enrolment respectively in models that only included data from the first survey). In an analysis of data from all surveys, negative binomial regression with mixed effects better explained the distribution of mosquito bite counts than mixed effects Poisson models (p<0.001), after adjustments for age, which influences *Anopheles* exposure in this population (*Gonçalves et al., 2017*), survey and intra-household data correlation. In sensitivity analyses that assigned blood meals with multiple human DNA sources to the least exposed individuals in each household, the ~20% of the population with most mosquito bites were linked to at least 70% of matched blood meals (*Figure 1—figure supplement 1*). We also performed an analysis that assigned multiple source blood meals to study participants based on minimal genetic distances to those meals (*Gonçalves et al., 2017*). This approach results in a similar pattern:~20% of the population received 78.6, 76.3% and 94.2% of all mosquito bites at the start, peak and end of the transmission season, respectively.

For individuals who participated in all surveys, there were positive correlations between numbers of matched mosquitoes (1) at the start and peak of transmission season (Spearman's rank correlation coefficient 0.24, p=0.04) and (2) at the start and end of transmission season (Spearman's rank correlation coefficient 0.34, p=0.002), suggesting some consistency in preferential biting. However, some individuals with highest numbers of matched blood meals at the peak of the transmission season received few or no mosquito bites in other surveys (*Figure 2a*).

Within-household heterogeneity in mosquito exposure was observed (*Figure 2b*): considering data from houses with at least three study participants and five matched mosquitoes in single surveys, in 4/8, 14/15 and 7/7 households at the start, peak and end of the transmission season

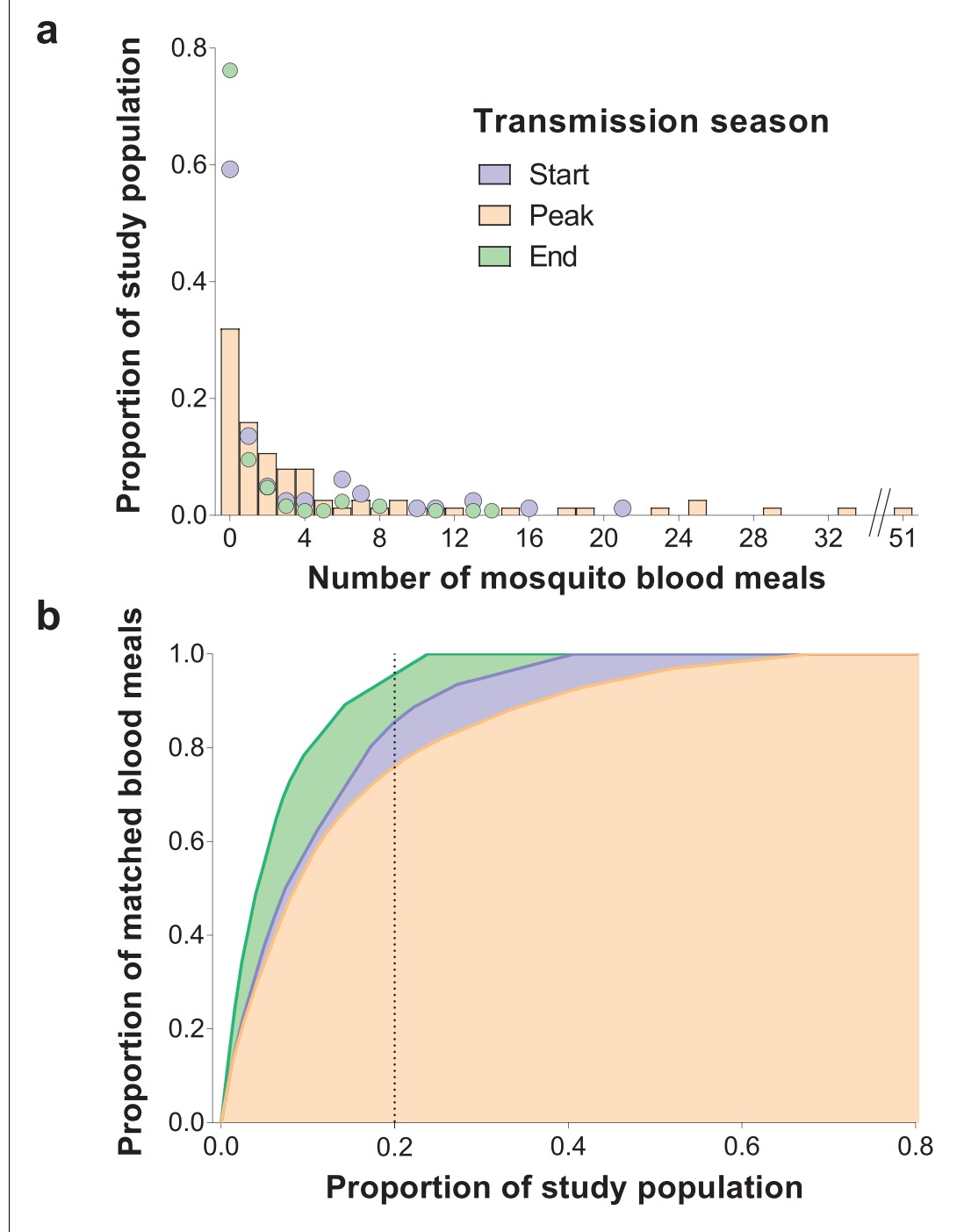

**Figure 1.** Inter-individual variation in exposure to *Anopheles* mosquitoes. In (**a**) the frequency distributions of mosquito blood meals matched to each study participant are presented for the three different surveys. Data from the peak transmission season are presented as bars; data from the other surveys (non-zero proportions) are presented as coloured circles. In (**b**) the cumulative proportion of mosquito blood meals (y-axis) matched to study participants (x-axis) sorted by number of mosquito bites received is presented for the different surveys. The points where the dotted vertical line intersects the three curves correspond to the highest proportions of mosquito blood meals linked to 20% of the study population. At the peak of the transmission season, six individuals were absent or only present during one collection day and were not included in this graph. Only singly matched bloodfed mosquitoes linked to individuals living in the same household where they were collected are included in this figure.

DOI: https://doi.org/10.7554/eLife.32625.003

The following source data and figure supplements are available for figure 1:

**Source data 1.** Distribution of mosquito blood meals matched to study participants by survey.

*Figure 1 continued on next page*

*Figure 1 continued*

DOI: https://doi.org/10.7554/eLife.32625.004

**Figure supplement 1.** Sensitivity analyses.

DOI: https://doi.org/10.7554/eLife.32625.005

**Figure supplement 2.** Cumulative proportion of *Anopheles* species-specific blood meals (y-axes) matched to study participants (x-axes) sorted by number of mosquito bites received.

DOI: https://doi.org/10.7554/eLife.32625.006

**Figure supplement 3.** Correlations between vector species-specific numbers of matched blood meals.

DOI: https://doi.org/10.7554/eLife.32625.007

respectively, the most exposed individual was the source of at least 50% of matched mosquito blood meals. While individuals with high numbers of matched blood meals often lived in households with high total numbers of matched mosquitoes (*Figure 2—figure supplement 1*), in all houses included in this study there were individuals with relatively low mosquito exposure.

## Anopheles species-specific feeding choices

Mosquitoes collected during the study were genotyped for species identification. At the start of the transmission season, *Anopheles coluzzii* represented 44.7% (142/318) of all bloodfed mosquitoes, while at the peak and end of the transmission season most bloodfed mosquitoes were *Anopheles*

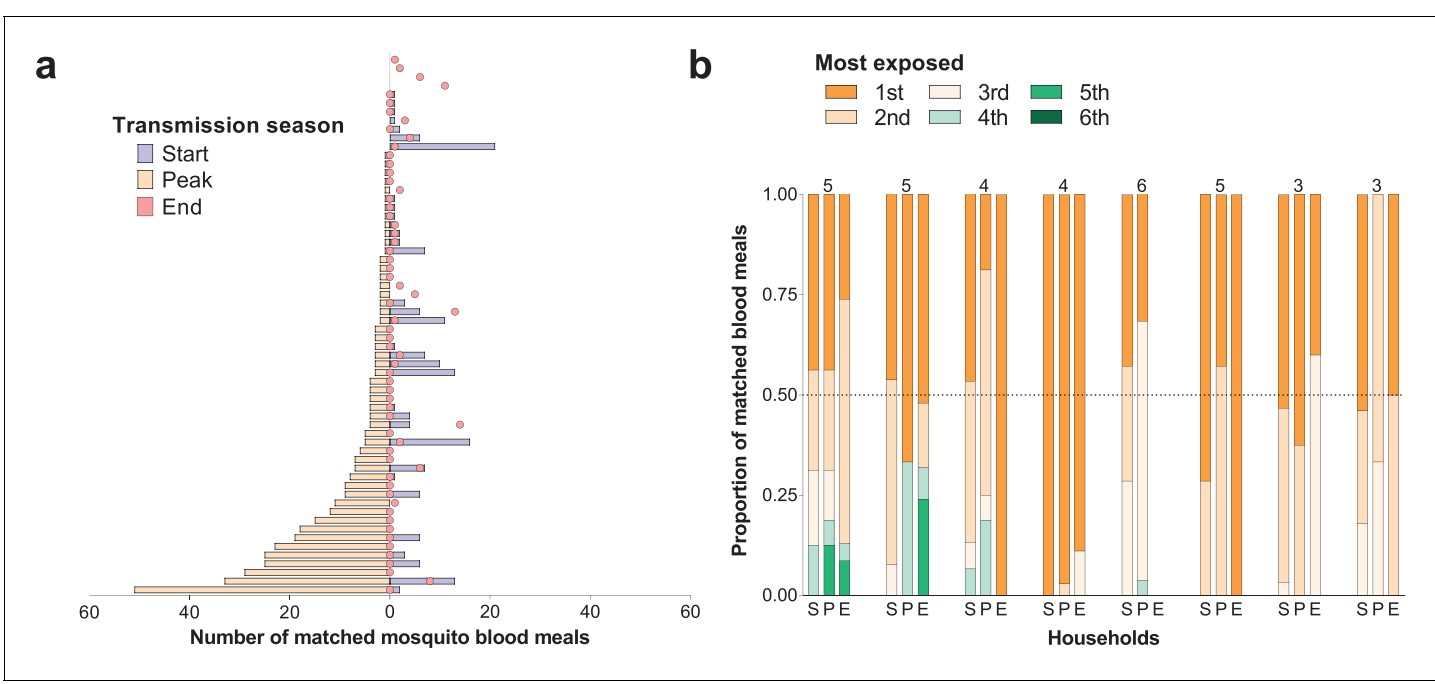

**Figure 2.** Temporal (**a**) and within-household (**b**) variation in exposure to malaria vectors. In (**a**) individuals were ranked (y-axis) according to the number of matched mosquito blood meals at the peak of the transmission season. The left and right, symmetrical, x-axes represent the number of mosquito bites each individual received at the peak (orange bars), and at the start (blue bars) and end (red circles) of the transmission season respectively. Only individuals present during at least three collection days per survey in all surveys and matched to at least one blood meal (N = 62) are included in this panel. In (**b**) each of the three columns corresponds to a different survey (S, Start of transmission season; P, Peak of transmission season; E, End of transmission season) for a select number of households. Individuals in the same household are denoted by different colours, which are consistent in the different surveys. The proportions of matched blood meals linked to each individual by household and survey are on the y-axis; only the eight households with at least five matched mosquitoes at the start of the transmission season and three or more study participants are shown. The numbers of individuals living in the households are shown above the columns.

DOI: https://doi.org/10.7554/eLife.32625.008

The following figure supplement is available for figure 2:

**Figure supplement 1.** Relationship between household- and individual-level exposures to *Anopheles* mosquitoes.

DOI: https://doi.org/10.7554/eLife.32625.009

*gambiae sensu stricto* (74.0 [450/608] and 53.7% [102/190], respectively). In all surveys, *A. coluzzii* mosquitoes had higher percentages of blood meals from multiple human sources (20.6, 22.0% and 18.9% at the start, peak and end of the transmission season, respectively) compared to *A. gambiae s. s.* mosquitoes (11.9, 16.1% and 10.3%). Species-specific distributions of blood meals singly matched to study participants are presented in *Figure 1—figure supplement 2* and suggest that heterogeneity in exposure to anopheline mosquitoes occurs irrespective of vector species. Of note, negative binomial models better explained the distribution of *Anopheles* species-specific mosquito bites compared to Poisson models, providing evidence of overdispersion that is not explained by covariates. In these models, the conditional overdispersion parameter was significantly different from zero: 1.9 (95% CI, 1.2–3.2) and 2.5 (95% CI, 1.8–3.6) for models of *A. coluzzii* and *A. gambiae s. s.* mosquito bites, respectively. Additionally, rates at which individuals were bitten by mosquitoes of different species were positively associated (*Figure 1—figure supplement 3*) (all p<0.001 in mixed effects negative binomial models that included number of *A. coluzzii* or *A. gambiae s. s.* matched blood meals, age and survey as covariates).

## Exposure to infected mosquito bites

The prevalence of malaria parasites in bloodfed mosquitoes identified via PCR of head and thorax was higher at the end versus start and peak of the transmission season (23.4, 4.9 and 8.0%, respectively), and slightly higher in singly-matched mosquitoes compared to mosquitoes with multiple meal sources (*Table 2*). *A. coluzzii* mosquitoes were less often infected compared to *A. gambiae s. s.* (odds ratio 0.41 95% CI, 0.22–0.76 in a model that adjusted for timing of survey). Individual- and

**Table 2.** Prevalence of mosquito infection, determined by PCR performed using mosquitoes' head-thoraces, by blood meal source (**A**) and mosquito species (**B**), and results of mixed effects logistic model on mosquito infection status (**C**).

| A | Prevalence of mosquito infection by blood meal source | | |
| --- | --- | --- | --- |
| | *Singly-matched, % (n/N)* | *Non-matched % (n/N)* | *Multiple human sources, % (n/N)* |
| Timing (Transmission Season) | | | |
| Start | 6.2 (10/160) | 4.5 (2/44) | 3.3 (3/91) |
| Peak | 9.9 (37/375) | 3.3 (2/61) | 0 (0/41) |
| End | 23.1 (25/108) | 25.0 (7/28) | 21.1 (4/19) |
| B | Prevalence of mosquito infection by mosquito species* | | |
| | *A. gambiae s. s., % (n/N)* | *A. coluzzii, % (n/N)* | *A. arabiensis, % (n/N)* |
| Timing (Transmission Season) | | | |
| Start | 4.9 (5/101) | 3.8 (4/106) | 9.7 (3/31) |
| Peak | 9.3 (36/386) | 2.5 (3/118) | 12.5 (1/8) |
| End | 31.4 (27/86) | 15.4 (8/52) | 0 (0/11) |
| C | Mixed effects logistic model on infection status* | | |
| | Odds ratio (95% CI) | P-value | |
| Mosquito species | | | |
| A. gambiae s. s. | Reference | | |
| A. coluzzii | 0.41 (0.22–0.76) | 0.005 | |
| A. arabiensis | 0.60 (0.19–1.84) | 0.37 | |
| Timing | | | |
| (Transmission Season) | | | |
| End | Reference | | |
| Start | 0.20 (0.09–0.45) | <0.001 | |
| Peak | 0.22 (0.11–0.44) | <0.001 | |

*Only mosquitoes with amplified human DNA were included in these calculations

DOI: https://doi.org/10.7554/eLife.32625.010

household-level frequencies of exposure to infected mosquitoes are presented in *Figure 3a*. Only 7.7, 23.6% and 10.5% of study subjects were linked to one or more potentially infectious mosquito meals at the start, peak and end of transmission season, respectively. In each survey, the two individuals with the highest numbers of matched meals from infected mosquitoes experienced 44–50% of all exposure to infected mosquitoes. As expected, there was a positive association between the number of potentially infective mosquito bites an individual received and the total number of

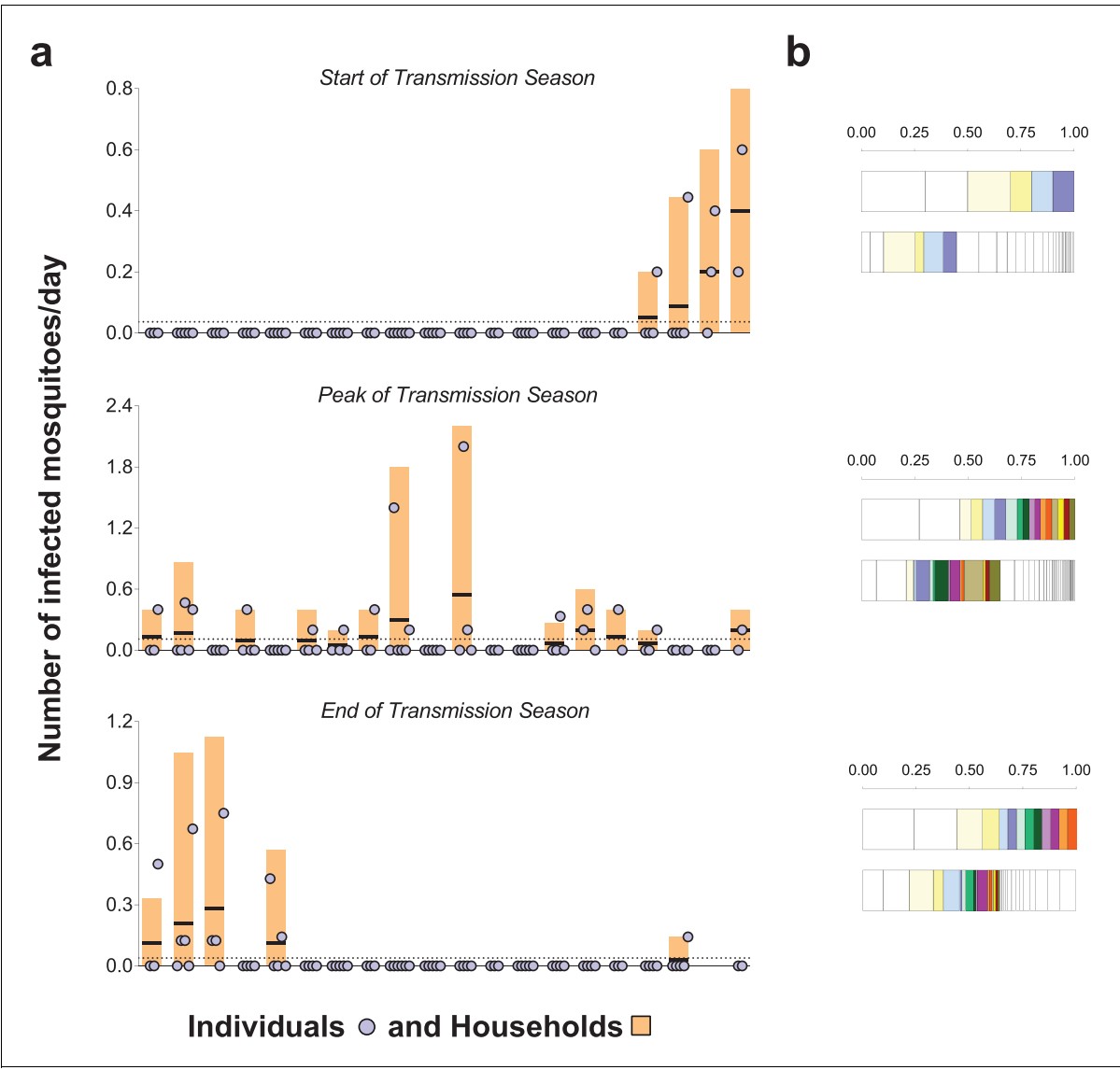

**Figure 3.** Rates of exposure to potentially infective mosquito bites (i.e., bites from mosquitoes with malaria parasites detected in their head-thoraces). (a) and their association with total mosquito exposure (b). In (a) blue circles represent study participants; individuals living in the same study house are presented together (orange bars). Y-axes show (i) the numbers of falciparum-positive mosquito bites per individual per day (blue circles), (ii) the total number of falciparum-positive fed mosquitoes collected in each house per collection day (orange bars), and (iii) the number of falciparum-positive fed mosquitoes collected in each house per individual resident in the house per collection day (black horizontal lines in orange bars); y-axes' limits vary to improve visualization. Horizontal dotted lines represent the average of iii over all houses. Only the 20 houses included in all three surveys are represented in this figure. In the three graphs, houses were ordered according to the number of infected fed mosquitoes collected at the start of the transmission season. In (b) in each pair of bars, each colour represents an individual: the top bar represents individuals matched to infected mosquitoes; the bottom bar, individuals matched to mosquitoes regardless of sporozoite status. Individuals who were only matched to uninfected mosquitoes are represented by white segments. The horizontal axes represent proportions of blood meals. Multiple source meals are not included in panel **b**.

DOI: https://doi.org/10.7554/eLife.32625.011

matched meals regardless of mosquito parasitological status (*Figure 3b*, p<0.001 in a mixed effects negative binomial model that had the total number of matched blood meals as a covariate and only included individuals with at least one matched meal). Conditional on the total number of bites, there was evidence of household-level clustering of infectious bites (intra-class [household] correlation 0.19, 95% CI 0.08–0.38, p<0.001 in a mixed effects logistic model that included singly matched mosquitoes); however, there was no evidence of individual-level clustering after adjustment for household-related variability (p=0.50).

### Effect of exposure heterogeneity on malaria $R_0$

The impact of biting heterogeneity on the resilience of vector-borne disease transmission to control interventions can be estimated using the methods described in (*Dietz, 1980*; *Dye and Hasibeder, 1986*): compared to uniform biting, the basic reproduction number ($R_0$) of a vector-borne disease, which corresponds to the number of secondary infections generated by an infectious individual in the absence of any control and any population immunity, is inflated by the factor $(1 + \alpha)$, where $\alpha$ is the squared coefficient of variation of the human biting rate. *Figure 4* illustrates the impact that more extreme heterogeneity can have on the calculated $R_0$ of malaria: $R_0$ is increased by a factor of 4.8, 4.2 and 8.4 in settings where mosquito exposure and its variability are similar to those observed at the start, peak and end of the transmission season in our study area. $R_0$ is high even when the skew is reduced by the conservative sensitivity analysis that allocates multiply fed mosquitoes' meals to the least exposed individuals, yielding $R_0$ increases by a factor of 3.3, 3.2 and 5.7 at the start, peak and end of the transmission season, respectively.

## Discussion

In this study, we quantified natural exposure to *Anopheles* mosquitoes using blood meals linked to household occupants. We observed significant differences in the numbers of bloodfed mosquitoes matched to study participants that are consistent with or even exceeding the Pareto rule, with ~20% of individuals being the source of more than 80% of all singly-matched blood meals. This heterogeneity was also apparent within-household, where the individual who received the most bites often contributed more than 50% of anopheline meals. Exposure to potentially infective mosquito bites also followed an aggregated pattern: 5.1, 13.9% and 6.5% of the population experienced 80% of parasite inoculations at the start, peak and end of the transmission season, respectively. Taken together, our observations indicate that relatively few individuals at different timepoints disproportionally contribute to malaria transmission by being repeatedly sampled and infected by malaria vectors. These data provide further insights into the mechanisms that lead to heterogeneity in human malaria infection risk (*Bejon et al., 2014*, *2010*; *Mwakalinga et al., 2016*; *Kangoye et al., 2016*).

Several previous studies have quantified inter-individual differences in attractiveness and exposure to mosquitoes using a range of methodologies. While studies involving experimental huts (*Lindsay et al., 1993*) and olfactometers (*Mukabana et al., 2002a*) demonstrated the influence of individual-level factors such as pregnancy (*Lindsay et al., 2000*) and infection status (*Lacroix et al., 2005*) on attractiveness to mosquitoes, community-wide assessments of wild-caught mosquitoes (*Port et al., 1980*; *Soremekun et al., 2004*; *Vazquez-Prokopec et al., 2016*) are necessary to quantify variation in exposure to malaria vectors over larger scales. A study in The Gambia (*Port et al., 1980*) in the 1980's used ABO group and haptoglobin typing to identify the sources of mosquito blood meals; the small number of variants in these markers (*Mukabana et al., 2002b*) limited the selection of households. In Tanzania (*Soremekun et al., 2004*), microsatellites were used to link blood meals to humans to assess the protection afforded by bed nets against mosquitoes; a high proportion of the ~250 analysed bloodfed mosquitoes were matched to individuals sleeping in the same room where they were collected, including in the village with bed nets, and 80% of matched blood meals came from less than 20% of the population. In our study, we matched 666 blood meals to individual study participants and observed that the distribution of mosquito bites was highly over-dispersed in relation to a Poisson assumption, with 76.0–95.5% of singly matched blood meals originating from ~20% of the study population. These results corroborate the findings of the study in Tanzania and of a different study in western Kenya (*Scott et al., 2006*), where 16% of the study participants were matched to 58% of *Anopheles* blood meals. By examining mosquito exposure repeatedly in an area of pronounced seasonality, we demonstrate even more unequal exposure patterns

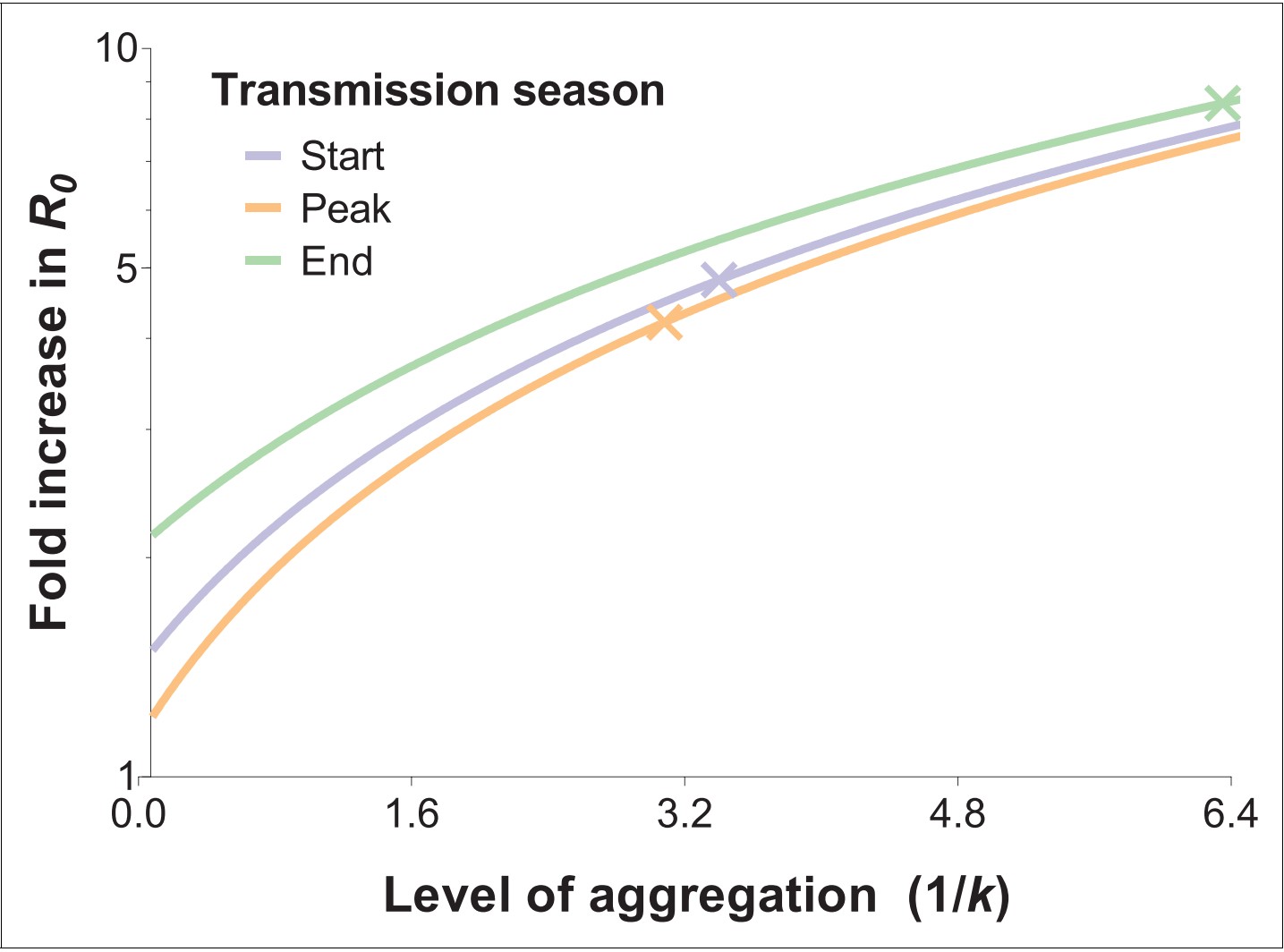

**Figure 4.** The relationship between the fold increase in $R_0$ and the level of aggregation in mosquito bites. This relation depends on the mean mosquito biting rate (see below), and the different curves represent the shape of this effect for different mosquito exposure levels: blue, orange and green lines correspond to settings where host-vector contact rates (mean numbers of matched mosquito bites) are similar to those observed at the start, peak and end of the transmission season, respectively. Aggregation level increases along with the inverse of the aggregation parameter $k$ from the negative binomial distribution, where $k$ can be calculated as the squared mean number of mosquito bites ($\mu^2$) divided by the variance in bite number ($\sigma^2$) minus the mean, that is $k = \frac{\mu^2}{(\sigma^2 - \mu)}$. $R_0$ is inflated by the factor $(1 + \alpha)$ whereby $\alpha = \left(\frac{\sigma}{\mu}\right)^2$; in other words, $\alpha$ is the squared coefficient of variation (**Smith et al., 2007**). In the special case whereby bites are perfectly homogenous, the coefficient of variation is zero, and the standard formulation for $R_0$ is regained (**Woolhouse et al., 1991**; **Barbour, 1978**). The fold increases in $R_0$ for the levels of aggregation observed in our surveys are represented by the coloured crossed-lines.
DOI: https://doi.org/10.7554/eLife.32625.012

across three timepoints in the season. Applying simple, well-established (**Smith et al., 2007**; **Churcher et al., 2015**) methods, we estimated that the observed degree of mosquito biting heterogeneity could be linked to a 3-fold or higher increase in malaria $R_0$ compared to a random-mixing system. This suggests that in settings with considerable variation in mosquito exposure, heterogeneity in biting is likely to be a major determinant of the coverage of vector control interventions required to reduce and interrupt transmission (**Smith et al., 2007**). These $R_0$-focused calculations however have limitations. Firstly, although there is a positive relationship between $R_0$ and variability in *Anopheles* biting, in finite populations increasing aggregation in exposure does not necessarily equate to increasing community-wide infection burden, since in areas with highly heterogeneous mosquito biting a high proportion of parasite inoculations will be on already-infected individuals.

Additionally, the framework used here does not account for previously observed interactions between frequency of exposure to infected mosquitoes and rates of immunity acquisition (*Bejon et al., 2014*; *Bejon et al., 2010*; *Clarke et al., 2002*), which could influence blood stage parasite burden and potentially the production of transmission stages in highly exposed, highly immune individuals. Development of detailed mathematical models that explicitly incorporate immunity and spatial dependencies (see next paragraph) would allow more precise estimation of the long-term influence of the observed variation in mosquito exposure on malaria transmission dynamics.

The fact that the majority of human-mosquito encounters in each survey involved only a few individuals is a consequence of both between- and within-household variation in exposure. Indeed, at the peak of the transmission season, the household with the highest number of fed mosquitoes had ~150 times more than the household with the lowest number. Household characteristics such as construction material, number of windows, eaves as well as geographical proximity to mosquito breeding sites have all been previously associated with increased exposure to mosquitoes indoors and explain some of the between-household variation in mosquito abundance. To understand the implications of our current findings for malaria control, data on geographical and temporal clustering of *Anopheles* exposure, malaria infection and disease are needed. A study in Kenya (*Bejon et al., 2014*) showed that foci of clinical malaria are unstable and that one-month surveillance data, for example, might only have predictive value of higher-than-average transmission over short periods of time. This is consistent with our observation that the ranking of households according to the number of infected mosquito bites per individual varies over time (*Figure 3*) and poses considerable challenges to sustainably targeting high exposure households for maximum community benefit.

We also found that household heterogeneity is compounded by inter-individual variation in frequency of sampling by mosquitoes for people in the same house. At the peak of the transmission season, the maximum difference in numbers of matched meals for participants in the same house ranged from 1 to 51, that is a difference of up to ~10 mosquito bites per day. Age and body size can partially explain differences in attractiveness (*Muirhead-Thomson, 1951*) yet we often noted age-matched individuals in the same house with dissimilar mosquito exposure. Other individual-level characteristics such as odours (*Mukabana et al., 2002a*; *Qiu et al., 2006*), (effective) use of protective measures or behaviour will be relevant; in our study, bed net use and gametocyte carriage were not significantly associated with the number of mosquito bites an individual received though this study was not designed to assess the influence of these factors on mosquito exposure. Of note, the age distribution of our study population (*Table 1*) might not reflect the true demography of the region. Whilst our results are in agreement with previous studies, we cannot exclude that a difference in age composition related to the fact that only houses with at least one child were included in the study may have influenced our heterogeneity estimates. Furthermore, we observed that blood meals from 8.2% of tested mosquitoes did not amplify for human DNA. Although it is possible that some of these mosquitoes fed on domestic animals, since it is unlikely that non-human biting would divert bites from specific hosts, we do not expect this to bias our individual-level estimates.

A technical limitation of this study was that our analyses only used data from ~70% of all successfully typed mosquito meals. Of those not matched, 139/292 were single blood meals from individuals not in our study houses and 153/292 had evidence of multiple blood meals. The distribution of these bites, each representing at least two host-vector encounters, is likely to be an important determinant of *Anopheles* exposure, in particular if some individuals are over-represented in these mixed meals, for example due to frequent defensive behaviour that leads to interrupted mosquito feeding and multiple probing. However, in a conservative sensitivity analysis that assigned these meals to the individuals with lowest exposure in each household, most mosquito bites were still linked to a small proportion of the population. Those mosquitoes whose blood meals did not match residents of study houses are likely to have fed in neighbouring houses or outdoors before entering study houses. The genetic profiles of these unmatched blood meals also suggest a heterogeneous feeding pattern: in the three houses with more than 10 unmatched mosquitoes, 16/41, 8/13 and 31/33 blood meals had the same profile. The only study house where a resident did not provide blood sample for matching had 8/10 unmatched meals presumably from the same human source. Concurrent collection of fed mosquitoes indoors and outdoors would help to understand how biting behaviour influences overall exposure.

Broad differences in anthropophily between mosquito species are well recognised (*Takken and Verhulst, 2013*) however there are fewer data (*Knols et al., 1995b*) on whether different *Anopheles*

species have different feeding preferences with regards to individual humans. In this study, we analysed blood meals in mosquitoes from three species: *A. gambiae s. s.*, *A. coluzzii* and *Anopheles arabiensis*. Although seasonal differences in species abundance were evident, we observed aggregation in human biting irrespective of vector species and exposure to one species was positively associated with exposure to the others. Though we only collected endophillic mosquitoes, it suggests that where vectors are anthropophilic heterogeneity in exposure to anopheline mosquitoes is a common epidemiological phenomenon including in areas with mixed vector species. Additionally, we observed that the most numerous species in our study area, *A. coluzzii* and *A. gambiae s. s.*, differed in two key parameters that influence the transmission potential of mosquitoes: the likelihood of feeding on multiple individuals and the prevalence of sporozoites. There is evidence from membrane feeding experiments performed in Senegal (*Ndiath et al., 2011*) that *A. gambiae s. s.* mosquitoes might be more susceptible to malaria infection, although this association was not observed in another transmission study (*Gnémé et al., 2013*). Another possible explanation for the observed species-related difference in sporozoite prevalence could be variation in mosquito survival and consequently age structure. The higher frequency of falciparum infection in *A. gambiae s. s.* mosquitoes confirms, together with its relatively high abundance, the prime importance of this species for malaria transmission in the study setting. The contributions of the different *Anopheles* species to the mosquito infectious reservoir also depend on the number of potential parasite inoculations per mosquito-time: mosquitoes with multiple blood meals are more likely to infect more than one individual on a single night compared to mosquitoes with single-source blood meals, assuming sufficient quantities of sporozoites are inoculated during the probing of the different hosts. The higher proportion of blood meals with multiple human DNA sources suggests that *A. coluzzii* mosquitoes also contribute significantly to the incidence of infection in humans. Importantly, as species composition of local vector populations varies in the course of the rainy season (*Dao et al., 2014*), these differences might impact the rate of infection propagation in human populations over the course of a single transmission season as relative abundances of the different vector species change.

We also determined how the frequency of host-vector contacts might influence malaria infection risk in human populations. In our study area, there was considerable variation in household-specific exposure to sporozoite-positive mosquitoes (*Figure 3*). At the individual level, a few individuals received multiple potentially infective bites while between 75% and 90% of the study participants were not matched to feeding by infected mosquitoes during the three study periods. Whilst this does not represent all infected mosquito bites these individuals receive during an entire transmission season, it does highlight the degree of heterogeneity in likely parasite inoculations and the limitations of using population- or region-wide entomological measures of transmission that do not capture this small-scale variation. These results also suggest that the use of blood meal genotyping with concurrent assessment of mosquito sporozoite carriage during epidemiological studies could improve our understanding of the heterogeneity in clinical malaria risk (*Mwangi et al., 2008*; *Loucoubar et al., 2013*; *Ndungu et al., 2015*). Furthermore, whilst we observed a positive association between the total number of mosquito bites and the number of bites from sporozoite infected mosquitoes at the individual level, our data suggest clustering of sporozoite-positive mosquito bites at the household level that was not related to local mosquito abundance. This phenomenon could be linked to (i) a correlation between prevalence of infection in mosquitoes and in humans in the same households assuming limited mixing of mosquitoes and humans (e.g. due to the presence of breeding sites near houses, reducing the distances travelled by mosquitoes between consecutive blood meals [*Le Menach et al., 2005*], or to feeding site-fidelity in *Anopheles* mosquitoes [*McCall et al., 2001*]) or possibly to (ii) factors which affect mosquito survival on a very local scale, since mosquito age is associated with cumulative risk of sporozoite infection (*Lines et al., 1991*).

In summary, although studies have assessed natural exposure to vectors of other infections, such as *Aedes* (*Harrington et al., 2014*; *Liebman et al., 2014*) and *Culex* (*Michael et al., 2001*), only limited data are available for *Anopheles* mosquitoes (*Soremekun et al., 2004*; *Scott et al., 2006*). In our field site, characterized by high malaria transmission intensity, we show significant heterogeneity both between and within households in terms of the number of mosquito blood meals and the distribution of potentially infectious mosquito feedings; these patterns are consistent with the 20/80 rule and support the design of interventions that aim to reduce transmission by targeting a small proportion of the population. Opportunities to specifically target high-exposure households depend on the

operational feasibility to identify these households and the stability in exposure over time. A quantitative understanding of the processes leading to heterogeneity in mosquito exposure and its temporal variability would inform at which level such interventions may be targeted in different settings; this would require quantification of the relative contributions of household-level factors, differential attractiveness to mosquitoes and human behavioural factors.

## Materials and methods

### Study area and mosquito collections

This study was conducted in Balonghin (health district of Saponé, Burkina Faso). The main economic activities in this area are subsistence farming and keeping livestock . Malaria transmission is seasonal, occurring between August and December following rainfall between June and September (*Ouédraogo et al., 2013*), and falciparum parasite prevalence is high, above 80% by *18S* qPCR, during the transmission season (*Gonçalves et al., 2017*). At the time of this study, seasonal malaria chemoprophylaxis was not part of national guidelines in Burkina Faso. Indoor mosquito collections were performed at the end of the 2013 transmission season (October – December), and at the start (June – July 2014) and peak (September 2014) of the following wet season. Every week mosquito collections were performed in five households. Forty households with at least one household member <15 years of age were included in the first survey. For each household, mosquitoes were collected between 7 and 9 AM by mouth aspiration from walls and ceilings for a maximum of 15 min per sampling morning for seven days or until 30 bloodfed *Anopheles* mosquitoes were collected. In 2014, mosquito collections were performed in 20 of these 40 initial households over 10 days (5 days in July, and five in September). Bloodfed mosquitoes had their head-thoraces stored for speciation and malaria infection assessment by PCR (see below), and their abdomens containing blood meal material squeezed onto filter paper that was stored with desiccant until DNA extraction and further analyses.

### Parasite detection in humans

Finger prick blood samples were collected at enrolment and stored in RNAprotect Cell Reagent for automatic extraction of total nucleic acid using a MagNAPure LC automatic extractor (Total Nucleic Acid Isolation Kit–High Performance; Roche Applied Science, Indianapolis, IN, USA). *18S* qPCR and *Pfs25* mRNA qRT-PCR were used to detect falciparum parasites and gametocytes, respectively, using established protocols (*Hermsen et al., 2001*; *Stone et al., 2017*).

### Mosquito blood meal typing

Genetic typing of blood meal samples has been described in detail elsewhere (*Gonçalves et al., 2017*). Briefly, bloodfed mosquitoes' abdomens were processed using Boom extraction method (*Boom et al., 1990*). The Authentifiler PCR Amplification kit (Applied Biosystems), with nine human microsatellite markers and one gender marker, was used to compare human DNA in blood meals and in blood samples collected from study participants. Capillary electrophoresis was used to determine DNA profiles. Mosquito blood meals with more than two alleles in at least three loci were considered to have multiple human sources.

### Mosquito speciation assay and malaria parasite detection by PCR

DNA was extracted from individual head-thoraces using the DNAzol procedure (Invitrogen, Cambridge, UK). Mosquito speciation was performed using a single PCR-RFLP assay as described by Fanello and colleagues (*Fanello et al., 2002*), and nested PCR (*Snounou et al., 1993*) was used for *Plasmodium falciparum* detection.

### Statistical analysis

Stata version 14 (Stata Corporation, College Station, TX, USA) was used for statistical analysis. Demographic data were not available for five houses where no bloodfed mosquitoes were collected; study participants living in these households are not included in the analyses presented here. We used the number of bloodfed *Anopheles* mosquitoes collected per day per household when comparing different surveys. Wilcoxon signed-rank test was used to assess within-household changes in

*Anopheles* abundance. To estimate household-specific exposure to infected mosquitoes (*Figure 3*), we only considered singly matched mosquitoes and mosquitoes with blood meals that had multiple human DNA sources, and excluded from these calculations mosquitoes whose blood meals did not match house residents; for individuals with at least one matched mosquito that was not tested for sporozoite status, the number of infected bites was estimated based on total number of matched meals and proportion of tested mosquitoes that were positive. A mixed effects logistic model was used to assess the association between mosquito falciparum infection status and mosquito species; the analysis was adjusted for survey time (fixed effect), and household of collection was used as random effect. To describe the temporal variation in relative abundance of the different *Anopheles* species, data from all mosquitoes with speciation results were used. In other analyses, that assessed *Anopheles* species-specific proportions of blood meals with multiple human sources and prevalences of sporozoites, only data from mosquitoes with both blood meal and head-thorax molecular assays were included (645/666 singly matched, 151/153 multiple, and 132/139 non-matched meals).

To generate *Figure 1b* and estimate the highest proportion of singly matched mosquito meals linked to 20% of study participants, only individuals who were present in at least three collection days were considered. Mixed effects negative binomial models (*Hilbe, 2011*; StataCorp *et al., 2013*) were used to assess overdispersion in exposure to mosquitoes, after adjustment for variability linked to age categories and survey time (fixed effects) and household-level variation (random effect). These models were also used in *Anopheles* species-specific analyses. Likelihood-ratio test compared mixed effects negative binomial models and mixed effects Poisson models.

## Ethics
Ethical clearance was obtained from the London School of Hygiene and Tropical Medicine ethics committee (reference number 6447), and the ethical review committee of the Ministry of Health of Burkina Faso (2013-7-58).

## Acknowledgements
We would like to thank the families who participated in this study. This work was supported by the Bill and Melinda Gates Foundation (AFIRM OPP1034789). TB is further supported by a fellowship of the Netherlands Organization for Scientific Research (VIDI fellowship grant number 016.158.306).

## Additional information

### Funding

| Funder | Grant reference number | Author |
| --- | --- | --- |
| Bill and Melinda Gates Foundation | AFIRM OPP1034789 | Teun Bousema<br>Chris Drakeley |
| Nederlandse Organisatie voor Wetenschappelijk Onderzoek | 016.158.306 | Teun Bousema |

The funders had no role in study design, data collection and interpretation, or the decision to submit the work for publication.

### Author contributions
Wamdaogo M Guelbéogo, Investigation, Methodology, Writing—original draft, Writing—review and editing; Bronner Pamplona Gonçalves, Formal analysis, Investigation, Visualization, Methodology, Writing—original draft, Writing—review and editing; Lynn Grignard, Samuel S Serme, Kjerstin Lanke, Soumanaba Zongo, Issiaka Soulama, Dimitri W Wangrawa, N'Falé Sagnon, Investigation, Writing—review and editing; John Bradley, Nuno Sepúlveda, Laith Yakob, Formal analysis, Writing—review and editing; Joel Hellewell, Formal analysis, Methodology, Writing—review and editing; Teun Bousema, Chris Drakeley, Conceptualization, Funding acquisition, Investigation, Methodology, Writing—original draft, Project administration, Writing—review and editing

## Author ORCIDs

John Bradley 🄳 http://orcid.org/0000-0002-9449-4608

Laith Yakob 🄳 http://orcid.org/0000-0001-8639-4511

Teun Bousema 🄳 https://orcid.org/0000-0003-2666-094X

Chris Drakeley 🄳 http://orcid.org/0000-0003-4863-075X

## Ethics

Human subjects: Written informed consent was obtained from study participants. Ethical clearance was obtained from the London School of Hygiene & Tropical Medicine ethics committee (reference number 6447), and the ethical review committee of the Ministry of Health of Burkina Faso (2013-7-58).

## Decision letter and Author response

Decision letter https://doi.org/10.7554/eLife.32625.015

Author response https://doi.org/10.7554/eLife.32625.016

## Additional files

**Supplementary files**

• Transparent reporting form

DOI: https://doi.org/10.7554/eLife.32625.013

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
