## [Decision Letter]

Thank you for submitting your article "Variation in natural exposure to *anopheles* mosquitoes and its effects on malaria transmission" for consideration by *eLife*. Your article has been reviewed by 3 peer reviewers, and the evaluation has been overseen by Marc Lipsitch as Reviewing Editor and Prabhat Jha as the Senior Editor. The following individuals involved in review of your submission have agreed to reveal their identity: Marc Lipsitch (Reviewer #1); William Hawley (Reviewer #2); Richard Paul (Reviewer #3).

The reviewers have discussed the reviews with one another and the Reviewing Editor has drafted this decision to help you prepare a revised submission.

Summary:

This is a well-written manuscript that quantifies the uneven distribution of bites by malaria vectors within and between households in a village in Burkina Faso. With the specific exceptions noted below it appears to be a well-done study that is one of only few to document this phenomenon in the field using a large survey and capturing the detail (vector species, infected vs. total bites, within and between households, demographic predictors). The phenomenon has been shown theoretically to be important, so documenting how large it is makes a significant contribution to the field, which will be interesting to anyone concerned about malaria and to others interested in the role of heterogeneity in infectious diseases.

Revisions below are requested to improve the understanding of readers (including non-specialists) of the context and precise implications of the finding.

Essential revisions:

1) The Discussion and some of the Results presentation should be modified to consider the following perspectives:

The authors cite compelling theoretical demonstrations that *R_0_* is massively inflated with this level of heterogeneity, compared to all behavior being average (random biting). It is not clear to me for an infection with slowly-developing immunity that *R_0_* is the best measure of impact. Suppose in the extreme case that all the mosquitoes exclusively bit 2 people of the 100 in a community. That would certainly make for a high *R_0_* and make control by say mass vaccination or mosquito control challenging (in proportion to *R_0_* inflation). It wouldn't however increase the burden of disease much and might decrease it, as these people would rapidly become immune, making bites on them less productive of gametocytes and further transmission (to the extent that immunity impacts transmission). Moreover (as in STI models) extreme specialization of this sort increases *R_0_* but not necessarily prevalence because some "nonspecialized" bloodfeeding is necessary to disperse infection into the wider community. This is an important limitation of the *R_0_*-focused presentation. This paper is not the place to develop the theory further, but I think emphasizing the impact on *R_0_* means the discursive parts of the manuscript should focus on the difficulty of control (which is proportional to *R_0_*) not "transmission" or "endemicity" generally, which does not follow in a simple way.

Another perspective on the same issue: It might be worthwhile to state that the absolute values of *R_0_*, as calculated, are notional in the sense that these are the values that would pertain in the absence of any control and any population immunity. In the case of Burkina, there's plenty of both control effort and immunity, so the actual *R_0_* is considerably less. The important lesson is that some means for better targeting control measures might be considered. Some of these are by now anodyne – we can't merely target children, but adults must be included in our efforts. We are already know that poor housing and poverty are associated with higher prevalence of malaria. While this paper very convincingly tells us that there is individual level variability in transmission – strongly associated with spatial factors – we don't yet know whether the observed spatial heterogeneity in transmission is temporally stable. If it isn't, then targeting of interventions will be hard to accomplish. But for the purposes of publication decision, this is not really of importance, as the paper presents splendid data confirming some long-held theoretical and field-based suspicions.

2) Figure 2. The reviewers had divergent opinions on the figures. Two of us found this figure in particular quite confusing. Another wrote "The figures are lovely, creative, and informative, once their code is understood." It is up to the authors to consider whether greater clarity can be achieved without sacrificing information or readability. Also it would be helpful to describe Figure 2 more explicitly in the text.

3) Figure 4. We don't understand how there are 3 different curves since 1/k should give a single curve. This must be because the fold increase depends on more than 1/k, and I just don't follow the figure legend, which should provide the formula.

4) “…in our study, bed net use did not significantly influence the number of mosquito bites an individual received (P = 0.72)…”. This result should not be presented for the first time in the Discussion. A p value with no mention of effect size or CIs is bad statistical practice. More generally, the list of factors considered as potential risk factors should be presented with univariate association measures provided (point and CI); as it is we don't know what the authors tried and only know that one association was null, but not whether it was convincingly close to null or just underpowered (narrow or wide CI respectively).

5) The fifth paragraph of the Discussion is hard to understand – could not identify the main point.

6) Subsection “Study area and mosquito collections”. We would like more information. I don't understand what was done. How were households chosen and individuals in them identified? Was sleeping in the household the night of the mosquito collection necessary for inclusion (if not, maybe some heterogeneity is just because someone wasn't there). What is the mosquito trapping protocol (I don't know about mouth aspiration but want more detail anyway about how long, where, what kind of traps etc.)?

7) The paper's penultimate sentence looms over this effort: "…support the design of interventions that aim to reduce transmission by a targeting a small proportion of the population." The theoretical literature already admonishes us to do this, but your data help us to grapple with the question – should we be targeting households or individuals? I think the data clearly support the former, and would suggest that your Discussion say so. Figure 3, Figure 2—figure supplement 1, and your regression analysis – "there was no evidence of individual level clustering after adjustment for household related variability (p = 0.50)” – all support the conclusion that the challenge to programs is to identify houses that attract the most mosquitoes, and not, thankfully, the most attractive individuals within houses. To gain this understanding, you advocate for "studies that combine mosquito blood meal typing and parasitological and clinical follow up"; in an earlier phrase "factors which affect mosquito survival on a very local scale" are mentioned. What about the distribution and productivity of larval habitats? In a paper about heterogeneity, this is an essential and extremely heterogeneous (spatially and temporally) fact of nature that should be considered if we are to figure out how to target the individuals living in high transmission houses. Even if you don't agree with my interpretation of your data, I think it would be worthwhile to address the 'what do we need to target? – individuals or households?' question more directly if the intent is to inform research of programmatic usefulness.

8) It would be helpful to summarize key features of malaria transmission in this area: what is the prevalence of parasitemia or other measures of transmission, and what is the nature and extent of interventions in place?

Optional major revision:

The most surprising absence, given that people were bled for human genotyping and given that LaCroix et al. (LaCroix et al., 2005) showed an increased attractiveness of parasite infected people to mosquitoes, is that there is no information given on the parasite prevalence in the population at an individual or even general level (how much malaria is there and when). The mosquitoes were examined for parasite sporozoite stages, but if the bloodmeals were squeezed out for typing and the abdomens remained, then surely the mosquito bloodmeal could have been checked for parasites.

---

## [Author Response]

1) The Discussion and some of the Results presentation should be modified to consider the following perspectives:The authors cite compelling theoretical demonstrations that R_0_ is massively inflated with this level of heterogeneity, compared to all behavior being average (random biting). It is not clear to me for an infection with slowly-developing immunity that R_0_ is the best measure of impact. Suppose in the extreme case that all the mosquitoes exclusively bit 2 people of the 100 in a community. That would certainly make for a high R_0_ and make control by say mass vaccination or mosquito control challenging (in proportion to R_0_ inflation). It wouldn't however increase the burden of disease much and might decrease it, as these people would rapidly become immune, making bites on them less productive of gametocytes and further transmission (to the extent that immunity impacts transmission). Moreover (as in STI models) extreme specialization of this sort increases R_0_ but not necessarily prevalence because some "nonspecialized" bloodfeeding is necessary to disperse infection into the wider community. This is an important limitation of the R_0_-focused presentation. This paper is not the place to develop the theory further, but I think emphasizing the impact on R_0_ means the discursive parts of the manuscript should focus on the difficulty of control (which is proportional to R_0_) not "transmission" or "endemicity" generally, which does not follow in a simple way.

We agree that the fold-changes in *R_0_*, presented in Figure 4, do not necessarily reflect the true relationship between heterogeneity in vector exposure and local human infection burden (prevalence or incidence). We would also concur that highly heterogenous transmission can result in an extreme scenario where the infection burden is concentrated in, or even confined to, a very small proportion of the population. Therefore, *R_0_* is more representative of the coverage level of non-targeted vector control interventions required to interrupt transmission.

To clarify this and the subsequent comment, we have modified the following paragraph in the Discussion section:

“Applying simple, well-established (Smith et al., 2007; Churcher, Trape and Cohuet, 2015) methods, we estimated that the observed degree of mosquito biting heterogeneity could be linked to a 3-fold or higher increase in malaria *R_0_* compared to a random-mixing system. […] Development of detailed mathematical models that explicitly incorporate immunity and spatial dependencies (see next paragraph) would allow more precise estimation of the long-term influence of the observed variation in mosquito exposure on malaria transmission dynamics.”

We have also replaced the title of the subsection “Effect of exposure heterogeneity on transmission” (Results section) with “Effect of exposure heterogeneity on malaria *R_0_*”, and modified its contents (some of the modifications in this paragraph relate to other comments):

“The impact of biting heterogeneity on the resilience of vector-borne disease transmission to control interventions can be estimated using the methods described in (Dietz, 1980; Dye and Hasibeder, 1986): compared to uniform biting, the basic reproduction number (*R_0_*) of a vector-borne disease, which corresponds to the number of secondary infections generated by an infectious individual in the absence of any control and any population immunity, is inflated by the factor (1 + α), where α is the squared coefficient of variation of the human biting rate. […] *R_0_* is high even when the skew is reduced by the conservative sensitivity analysis that allocates multiply fed mosquitoes’ meals to the least exposed individuals, yielding *R_0_* increases by a factor of 3.3, 3.2 and 5.7 at the start, peak and end of the transmission season, respectively.”

Another perspective on the same issue: It might be worthwhile to state that the absolute values of R_0_, as calculated, are notional in the sense that these are the values that would pertain in the absence of any control and any population immunity. In the case of Burkina, there's plenty of both control effort and immunity, so the actual R_0_ is considerably less. The important lesson is that some means for better targeting control measures might be considered. Some of these are by now anodyne – we can't merely target children, but adults must be included in our efforts. We are already know that poor housing and poverty are associated with higher prevalence of malaria. While this paper very convincingly tells us that there is individual level variability in transmission – strongly associated with spatial factors – we don't yet know whether the observed spatial heterogeneity in transmission is temporally stable. If it isn't, then targeting of interventions will be hard to accomplish. But for the purposes of publication decision, this is not really of importance, as the paper presents splendid data confirming some long-held theoretical and field-based suspicions.

We agree that absolute values of *R_0_* should be interpreted in the context of interventions that reduce the actual reproduction number. Because of this obvious discordance, we did not estimate local malaria *R_0_*, opting instead to present changes in *R_0_* (relative to a homogenous-biting system) estimated from the new data on biting variation.

As noted in the response to the previous comment we have revised the Discussion to point out 1) that *R_0_* calculations concern scenarios where no control measures are in place in a population of immunologically naïve individuals, 2) that malaria *R_0_* might not be the best measure to assess the impact of biting heterogeneity on local malaria burden, 3) that *R_0_* amplification due to variation in mosquito exposure determines requirements for control strategies to be effective, and 4) that more complex mathematical models that account for correlations between malaria exposure and rate of immunity acquisition might inform how the distribution of mosquito bites influences different human malaria metrics.

In addition to these modifications related to *R_0_*, we also agree with the reviewers that characterising the temporal variation in the spatial distribution of mosquito bites is essential to determine the feasibility of spatially targeted interventions. Bejon and colleagues showed that foci of clinical malaria are unstable and that one-month surveillance data might only have predictive value of future hotspots for short periods of time (A micro-epidemiological analysis of febrile malaria in Coastal Kenya showing hotspots within hotspots. *eLife* 2014). This is consistent with our observation that the ranking of houses according to the number of infected mosquito bites per individual varies over time (Figure 3). To this, we add that determining the spatial scale of malaria transmission is also important. The average distance between locations where consecutive blood meals of the same mosquito take place is likely to influence the contribution of different households to community-wide transmission, and consequently is a measure of the maximal efficacy of spatially targeted interventions (see Perkins et al. Heterogeneity, Mixing, and the Spatial Scales of Mosquito-Borne Pathogen Transmission. PLOS Computational Biology 2013 for a detailed discussion on spatial scale of malaria transmission; see McCall et al. Evidence for memorized site-fidelity in *Anopheles arabiensis*. Trans R Soc Trop Med Hyg. 2001 and Le Menach et al. The unexpected importance of mosquito oviposition behaviour for malaria: non-productive larval habitats can be sources for malaria transmission. Malaria Journal 2005 for possible mechanisms influencing the scale of transmission).

We modified the following paragraph to emphasize the need to quantify temporal variation in the spatial distribution of mosquito bites. These changes are equally relevant to comment 7.

“The fact that the majority of human-mosquito encounters in each survey involved only a few individuals is a consequence of both between- and within-household variation in exposure. […] We also found that household heterogeneity is compounded by inter-individual variation in frequency of sampling by mosquitoes for people in the same house.”

2) Figure 2. The reviewers had divergent opinions on the figures. Two of us found this figure in particular quite confusing. Another wrote "The figures are lovely, creative, and informative, once their code is understood." It is up to the authors to consider whether greater clarity can be achieved without sacrificing information or readability. Also it would be helpful to describe Figure 2 more explicitly in the text.

We have now modified Figure 2 to present temporal variation in exposure in one single panel. This information is now represented in linear scale. We have also modified the Results section and the Materials and methods section where the calculations used to create the previous panels A and B were described.

“Figure 2. Temporal (A) and within-household (B) variation in exposure to malaria vectors. […] The numbers of individuals living in the households are shown above the columns.”

“However, some individuals with highest numbers of matched blood meals at the peak of the transmission season received few or no mosquito bites in other surveys (Figure 2).”

“Within-household heterogeneity in mosquito exposure was observed (Figure 2): considering data from houses with at least three study participants and five matched mosquitoes in single surveys, in 4/8, 14/15 and 7/7 households at the start, peak and end of the transmission season respectively, the most exposed individual was the source of at least 50% of matched mosquito blood meals.”

3) Figure 4. We don't understand how there are 3 different curves since 1/k should give a single curve. This must be because the fold increase depends on more than 1/k, and I just don't follow the figure legend, which should provide the formula.

The different curves in Figure 4 represent the effect of aggregation (*1/k*) on malaria *R_0_* in areas with mosquito exposure levels (i.e. average number of mosquito bites) similar to those observed in our study (i.e. start [blue], peak [orange] and end [green] of transmission season). These calculations are not intended to reflect seasonal biting variation, but rather to provide estimates of *R_0_* amplification in theoretical settings with vector densities and biting variation similar to those observed at different times in our study area. To clarify this, we have modified the legend of Figure 4 and the Results section (see below).

Additionally, when addressing this comment, we found an error in the calculations of the *R_0_* inflations. Essentially, the coefficient of variation was miscalculated as the variance divided by the mean bite rate when it should have been the standard deviation divided by the mean. While the biting variation recorded in the field still inflates *R_0_* considerably, the inflations are attenuated (now resulting in 4 to 8-fold increases in *R_0_* relative to homogenous mixing). We have corrected the sub-section “Effect of exposure heterogeneity on malaria *R_0_*” in the Results section to reflect this change. These do not alter the overall message of the manuscript or the large majority of the presented results. The corrected *R_0_* calculation resulted in changes in the last paragraph of the Results section and one paragraph in the Discussion section, the actual presentation of all entomological data is unaltered and correct.

*“*Figure 4. The relationship between the fold increase in *R_0_* and the level of aggregation in mosquito bites. […] The fold increases in *R_0_* for the levels of aggregation observed in our surveys are represented by the coloured crossed-lines.”

“The impact of biting heterogeneity on the resilience of vector-borne disease transmission to control interventions can be estimated using the methods described in (Dietz, 1980; Dye and Hasibeder, 1986): compared to uniform biting, the basic reproduction number (*R_0_*) of a vector-borne disease, which corresponds to the number of secondary infections generated by an infectious individual in the absence of any control and any population immunity, is inflated by the factor (1 + α), where α is the squared coefficient of variation of the human biting rate. […] *R_0_* is high even when the skew is reduced by the conservative sensitivity analysis that allocates multiply fed mosquitoes’ meals to the least exposed individuals, yielding *R_0_* increases by a factor of 3.3, 3.2 and 5.7 at the start, peak and end of the transmission season, respectively.”

*4)* “… in our study, bed net use did not significantly influence the number of mosquito bites an individual received (P = 0.72)…”*. This result should not be presented for the first time in the Discussion. A p value with no mention of effect size or CIs is bad statistical practice. More generally, the list of factors considered as potential risk factors should be presented with univariate association measures provided (point and CI); as it is we don't know what the authors tried and only know that one association was null, but not whether it was convincingly close to null or just underpowered (narrow or wide CI respectively).*

We previously showed that age influences mosquito exposure in this area (Gonçalves et al. Examining the human infectious reservoir for *Plasmodium falciparum* malaria in areas of differing transmission intensity. Nature Communications 2017). Here, we also assessed the effect of bed net use on mosquito exposure, although as we mentioned in the Discussion section, the study was not designed for this. We modified the Results section to describe the analysis that includes bed net use. Additionally, to address another comment (see “Optional major revision”), we have now included a statement on the effect of parasite carriage on mosquito biting.

“Throughout the study, a small number of individuals, mostly adults, were matched to considerably higher numbers of blood meals compared to the rest of the population. […] In an analysis of data from all surveys, negative binomial regression with mixed effects better explained the distribution of mosquito bite counts than mixed effects Poisson models (P < 0.001), after adjustments for age, which influences *Anopheles* exposure in this population (Goncalves et al., 2017), survey and intra-household data correlation.”

We have also modified the *Discussion section*:

“… in our study, bed net use and gametocyte carriage were not significantly associated with the number of mosquito bites an individual received though this study was not designed to assess the influence of these factors on mosquito exposure.”

5) The fifth paragraph of the Discussion is hard to understand – could not identify the main point.

In this paragraph, we discuss the effect of the heterogeneity described on the distribution of potentially infectious bites from mosquitoes. As with total mosquito counts, most sporozoite-positive mosquito blood meals were matched to a small proportion of participants. An unexpected finding was that infected bites clustered at the household level after adjustment for the number of mosquitoes collected. We believe this observation is important since it might be linked to previously unrecognized epidemiological phenomena. Indeed, possible explanations for this include 1) household-level differences in mosquito survival and average age, where mosquito age is associated with cumulative risk of infection (Lines et al. Human malaria infectiousness measured by age-specific sporozoite rates in *Anopheles gambiae* in Tanzania. Parasitology 1991), and 2) a time-lagged correlation between prevalences of infection in mosquitoes and in humans in corresponding households. Explanation 2 implies limited movement of mosquitoes in some areas, i.e. the distance travelled by mosquitoes between consecutive blood meals is small. This might occur for example in houses located near breeding sites, which allow oviposition after blood feeding and might increase the probability that mosquitoes feed in the same house afterwards. Another mechanism could involve feeding site-fidelity as previously described (McCall et al. Evidence for memorized site-fidelity in *Anopheles arabiensis*. Trans R Soc Trop Med Hyg. 2001).

We have improved the clarity of this paragraph. We have removed the statement that contrasted the proportion of the study population linked to potentially infective blood meals and the prevalence of infection in this population, since these two quantities are not directly comparable (we assessed exposure to infected mosquitoes during one week at the end of the transmission season, while infections in humans can last several months [Felger et al. The Dynamics of Natural *Plasmodium falciparum* Infections. PLoS ONE 2012]). We also included additional information about the clustering of infected mosquitoes at the household level.

“We also determined how the frequency of host-vector contacts might influence malaria infection risk in human populations. In our study area, there was considerable variation in household-specific exposure to sporozoite-positive mosquitoes (Figure 3). […] This phenomenon could be linked to i) a correlation between prevalence of infection in mosquitoes and in humans in the same households, assuming limited mixing of mosquitoes and humans (e.g. due to the presence of breeding sites near houses, reducing the distances travelled by mosquitoes between consecutive blood meals (Le Menach et al., 2005), or to feeding site-fidelity in *Anopheles* mosquitoes (McCall et al., 2001)), or possibly to ii) factors which affect mosquito survival on a very local scale, since mosquito age is associated with cumulative risk of sporozoite infection (Lines, Wilkes and Lyimo, 1991).

6) Subsection “Study area and mosquito collections”. We would like more information. I don't understand what was done. How were households chosen and individuals in them identified? Was sleeping in the household the night of the mosquito collection necessary for inclusion (if not, maybe some heterogeneity is just because someone wasn't there). What is the mosquito trapping protocol (I don't know about mouth aspiration but want more detail anyway about how long, where, what kind of traps etc.)?

Households with at least one child were eligible for inclusion in the first entomological survey. Residents in study houses were enrolled and asked to provide a finger prick blood sample that was used to match mosquito blood meals to humans. In the second and third surveys, twenty houses from the first survey with no changes in the number of residents (e.g., no births or deaths) were sampled again for bloodfed mosquitoes; priority was given to houses with highest mosquito exposure in the first survey. For the duration of catching in each survey we recorded whether each participant was present in the study house during the nights preceding mosquito collections. We used this information as exposure variable in statistical models of mosquito biting, including in the negative binomial models, and to exclude individuals with limited exposure data from some of the figures (Figure 1, Figure 2 and Figure 1—figure supplement 1).

We have now modified the Materials and methods section to give more details about mosquito collections:

“This study was conducted in Balonghin (health district of Saponé, Burkina Faso). […] Bloodfed mosquitoes had their head-thoraces stored for species confirmation and malaria infection assessment by PCR (see below), and their abdomens containing blood meal material squeezed onto filter paper that was stored with desiccant until DNA extraction and further analyses.”

7) The paper's penultimate sentence looms over this effort: "…support the design of interventions that aim to reduce transmission by a targeting a small proportion of the population." The theoretical literature already admonishes us to do this, but your data help us to grapple with the question – should we be targeting households or individuals? I think the data clearly support the former, and would suggest that your Discussion say so. Figure 3, Figure 2—figure supplement 1, and your regression analysis – "there was no evidence of individual level clustering after adjustment for household related variability (p = 0.50)” – all support the conclusion that the challenge to programs is to identify houses that attract the most mosquitoes, and not, thankfully, the most attractive individuals within houses. To gain this understanding, you advocate for "studies that combine mosquito blood meal typing and parasitological and clinical follow up"; in an earlier phrase "factors which affect mosquito survival on a very local scale" are mentioned. What about the distribution and productivity of larval habitats? In a paper about heterogeneity, this is an essential and extremely heterogeneous (spatially and temporally) fact of nature that should be considered if we are to figure out how to target the individuals living in high transmission houses. Even if you don't agree with my interpretation of your data, I think it would be worthwhile to address the 'what do we need to target? – individuals or households?' question more directly if the intent is to inform research of programmatic usefulness.

Indeed, our results indicate that exposure to infected mosquitoes is highly variable at the household level. In particular, at the start of the transmission season, a period which may determine the malaria burden during the subsequent transmission season, only a few houses had infected mosquitoes collected. As the reviewer suggests, targeting a proportion of households (and not targeting a proportion of individuals within households) could have a significant impact on potential transmission. The issue of the identification of households associated with high exposure to mosquitoes is obviously key. As mentioned by the reviewer, the spatial distribution of larval breeding sites is one likely factor determining spatial heterogeneity in transmission, as it determines both abundance and feeding after oviposition, which has an effect on the scale of transmission (see Le Menach et al. The unexpected importance of mosquito oviposition behaviour for malaria: non-productive larval habitats can be sources for malaria transmission. Malaria Journal 2005 for a detailed discussion). However, as several studies have shown, other factors such as household construction and adherence to existing vector control will also play roles in this and that identification of factors will be ecotype dependent. Whilst we did not find evidence for statistically significant associations between bed net use and vector exposure, it is an intuitively important factor in determining mosquito exposure and one we cannot rule out as determinant. Our response to comment 1 (modifications made in the third and fourth paragraphs of the Discussion) are relevant to this comment. We have also modified the final paragraph of the Discussion section:

“In summary, although studies have assessed natural exposure to vectors of other infections, such as Aedes (Harrington et al., 2014; Liebman et al., 2014) and Culex (Michael et al., 2001), only limited data are available for *Anopheles* mosquitoes (Soremekun et al., 2004, Scott et al., 2006). […] A quantitative understanding of the processes leading to heterogeneity in mosquito exposure and its temporal variability would inform at which level such interventions may be in different settings; this would require quantification of the relative contributions of household-level factors, differential attractiveness to mosquitoes and human behavioural factors.”

Of note, demographic groups are targeted in some settings primarily for morbidity reduction -SMC, IPTp,_I_ etc., though some of the data from SMC suggest there may also be a community effect on parasite prevalence in targeting children under 10. In a related analysis, we estimated the contribution of different age groups to transmission after accounting for the age structure of the population, relative infectivity and heterogeneity in exposure. Extending drug delivery to older groups via SMC or through schools represents a logistically attractive complimentary approach to vector control.

8) It would be helpful to summarize key features of malaria transmission in this area: what is the prevalence of parasitemia or other measures of transmission, and what is the nature and extent of interventions in place?

Data from a recent epidemiological study showed that the prevalence of falciparum parasites during malaria transmission season was 83.8% by molecular methods, which is similar to the prevalence of 79.2% during the first mosquito collection survey. While seasonal chemoprophylaxis was not recommended in Burkina Faso at the time of this study, reported bed net use was high (90.0%, 87.1% and 88.6% for children < 5 years, schoolchildren and adults, as we previously described [Gonçalves et al. Examining the human infectious reservoir for *Plasmodium falciparum* malaria in areas of differing transmission intensity. Nature Communications 2017]). We now mention this information in the Materials and methods and Results sections.

“This study was conducted in Balonghin (health district of Saponé, Burkina Faso). […] Bloodfed mosquitoes had their head-thoraces stored for speciation and malaria infection assessment by PCR (see below), and their abdomens containing blood meal material squeezed onto filter paper that was stored with desiccant until DNA extraction and further analyses.”

“We performed indoor resting collections of anopheline mosquitoes in an area with seasonal malaria transmission in Burkina Faso. […] Reported bed net use among the study participants was high (111/126, 88.1%).”

Optional major revision:The most surprising absence, given that people were bled for human genotyping and given that LaCroix et al. (LaCroix et al., 2005) showed an increased attractiveness of parasite infected people to mosquitoes, is that there is no information given on the parasite prevalence in the population at an individual or even general level (how much malaria is there and when). The mosquitoes were examined for parasite sporozoite stages, but if the bloodmeals were squeezed out for typing and the abdomens remained, then surely the mosquito bloodmeal could have been checked for parasites.

In response to the reviewers’ comment, we have now included parasitology data for the first season when samples were available for this purpose. We tested these human samples, stored in RNA protect cell reagent, for parasite and gametocyte detection using *18S* qPCR and *Pfs25* qRT-PCR and results are now included in the Results section and in Table 1. We observed high parasite and gametocyte prevalence in human samples and, potentially because of this, there was no association between mosquito exposure and parasite carriage in humans. We do agree that this is an interesting phenomenon and highly valuable to explore in the context of natural mosquito exposure. Given the estimated effect sizes of gametocyte carriage (~2-fold higher mosquito exposure; Busula et al. J Infect 2017; Lacroix et al.PLoS Biol 2005) and body surface area/age (up to 21-fold higher exposure, Gonçalves et al. Examining the human infectious reservoir for *Plasmodium falciparum* malaria in areas of differing transmission intensity. Nature Communications 2017), it may be challenging to reliably detect and quantify an impact of human parasite status on mosquito exposure in natural settings.

“We performed indoor resting collections of anopheline mosquitoes in an area with seasonal malaria transmission in Burkina Faso. […] We collected 325, 620 and 190 bloodfed *Anopheles* mosquitoes at the start (2014), peak (2014) and end (2013) of the transmission season, respectively (Table 1).”

“Throughout the study, a small number of individuals, mostly adults, were matched to considerably higher numbers of blood meals compared to the rest of the population. […] In an analysis of data from all surveys, negative binomial regression with mixed effects better explained the distribution of mosquito bite counts than mixed effects Poisson models (P < 0.001), after adjustments for age, which influences *Anopheles* exposure in this population (Goncalves et al., 2017), survey and intra-household data correlation.”

“… in our study, bed net use and gametocyte carriage were not significantly associated with the number of mosquito bites an individual received though this study was not designed to assess the influence of these factors on mosquito exposure.”

“Parasite detection in humans

Finger prick blood samples were collected at enrolment and stored in RNAprotect Cell Reagent for automatic extraction of total nucleic acid using a MagNAPure LC automatic extractor (Total Nucleic Acid Isolation Kit–High Performance; Roche Applied Science, Indianapolis, IN, USA). 18S qPCR and Pfs25 mRNA qRT-PCR were used to detect falciparum parasites, and gametocytes, respectively, using established protocols (Hermsen et al., 2001; Stone et al., 2017).”(New sub-section in the Materials and methods section.)